# Morphological and Quantitative Evidence for Altered Mesenchymal Stem Cell Remodeling of Collagen in an Oxidative Environment—Peculiar Effect of Epigallocatechin-3-Gallate

**DOI:** 10.3390/polym14193957

**Published:** 2022-09-22

**Authors:** Regina Komsa-Penkova, Svetoslava Stoycheva, Pencho Tonchev, Galya Stavreva, Svetla Todinova, Galya Georgieva, Adelina Yordanova, Stanimir Kyurkchiev, George Altankov

**Affiliations:** 1Department of Biochemistry, Medical University Pleven, 5800 Pleven, Bulgaria; 2Department of Surgery, Medical University Pleven, 5800 Pleven, Bulgaria; 3Department of Experimental and Clinical Pharmacology, Medical University Pleven, 5800 Pleven, Bulgaria; 4Institute for Biophysics and Biomedical Engineering, Bulgarian Academy of Sciences, 1330 Sofia, Bulgaria; 5Tissue Bank BulGen, Str. Hristo Blagoev 25, 1330 Sofia, Bulgaria; 6Associate Member of Institute for Biophysics and Biomedical Engineering, Bulgarian Academy of Sciences, 1330 Sofia, Bulgaria; 7Research Institute, Medical University Pleven, 5800 Pleven, Bulgaria

**Keywords:** adipose tissue-derived mesenchymal stem cell, ADMSC, collagen type I, EGCG, oxidation, remodeling

## Abstract

Mesenchymal stem cells (MSCs) are involved in the process of extracellular matrix (ECM) remodeling where collagens play a pivotal role. We recently demonstrated that the remodeling of adsorbed collagen type I might be disordered upon oxidation following its fate in the presence of human adipose-derived MSC (ADMSCs). With the present study we intended to learn more about the effect of polyphenolic antioxidant Epigallocatechin-3-gallate (EGCG), attempting to mimic the conditions of oxidative stress in vivo and its putative prevention by antioxidants. Collagen Type I was isolated from mouse tail tendon (MTC) and labelled with FITC before being oxidized according to Fe^2+^/H_2_O_2_ protocol. FITC-collagen remodeling by ADMSC was assessed morphologically before and after EGCG pretreatment and confirmed via detailed morphometric analysis measuring the anisotropy index (AI) and fluorescence intensity (FI) in selected regions of interest (ROI), namely: outside the cells, over the cells, and central (nuclear/perinuclear) region, whereas the pericellular proteolytic activity was measured by de-quenching fluorescent collagen probes (FRET effect). Here we provide morphological evidence that MTC undergoes significant reorganization by the adhering ADMSC and is accompanied by a substantial activation of pericellular proteolysis, and further confirm that both processes are suppressed upon collagen oxidation. An important observation was that this abrogated remodeling cannot be prevented by the EGCG pretreatment. Conversely, the detailed morphometric analysis showed that oxidized FITC-collagen tends to accumulate beneath cells and around cell nuclei, suggesting the activation of alternative routes for its removal, such as internalization and/or transcytosis. Morphometric analysis also revealed that both processes are supported by EGCG pretreatment.

## 1. Introduction

Collagens are the most abundant proteins of the extracellular matrix (ECM) and are important for the mechanical properties of connective tissues, regulation of cell function, and communication [1]. It recently became clear that they are also tightly involved in the regulation of adult stem cell functioning, such as renewal and differentiation [2]. Mesenchymal stem cells (MSCs) isolated from different sources can differentiate into other types of cells [3,4], providing unique properties for various cellular therapies. In humans, these sources include bone marrow, adipose, umbilical tissues, and amniotic fluid [2,5,6]. An important unresolved issue, however, is how to guide the behavior of MSCs in engineered environments.

Under physiological conditions, MSCs are capable of self-renewal and differentiation induced by environmental stimuli: soluble growth factors [7] and distinct structural cues related to the tissue specific ECM organization. Although it is well known that the ECM has important roles in regulating the development and homeostasis of eukaryotic cells [8,9], relatively little is known as to how stem cells behave within the specific ECM environment (often approximated as stem cells niche) [10,11]. The molecules that are usually associated with ECM include collagens, laminins, fibronectin, proteoglycans, and other specific glycoproteins [12], which are associated with the rough ECM [13], or highly specialized basement membranes (BM) [14,15]. The way they assemble determines the structural organization of the stem cells niche and regulates the balance between stem cell differentiation and self-renewal [16]. Adipose tissue-derived MSCs (ADMSCs) draw notable attention in tissue engineering, combining relatively easy availability, lessened donor site morbidity, and multi-potency [2,17]. In addition, like most MSCs, they lack expression of major histocompatibility complex (MHC) class II surface molecules (important for immune rejection) and further provide regulatory function, influencing the phenotype of immune cells and their cytokine secretions upon stimulation of their toll-like receptor (TLR) [18]. Thus, ADMSCs provide unique properties for various regenerative therapies.

Although our knowledge of the composition and function of the ECM molecules is continuously growing [19], structural aspects of the ECM, including the intimate organization of collagen molecules, are still not well understood, particularly in pathological environments [20]. Oxidative stress is one such condition that strongly affects the collagen structure and turnover [21], including synthesis, post-translational processing [22], and remodeling [23], but the effect on stem cell functioning is rather sparsely studied. In this respect, our recent study showing that the post translational oxidation of collagen type I (obtained from rat tails) significantly alters its remodeling by stem cells, including mechanical reorganization and proteolytic degradation [24], and opens the scores for further investigations.

The collagen superfamily in vertebrates comprises 28 members (I–XXVIII) with a common structural feature—the presence of a triple helix that can range between different types of collagens. For type I collagen, the most abundant type of collagen, it is about 96% [25], but in a tendon its content can reach about 100% of the total dry mass [26]. Thus, the typical structural “brand” of collagen type 1 is the tight right-handed triple helix composed of three left-handed polypeptide chains. The anisotropic character of collagen molecule and its sequence variability is widely studied and discussed. For example, it is well known that sequence variability along the length of collagen results in local differences in helix pitch, dynamics, and thermal stability [27], which suggests that distinct regions of the triple helix may be sensitive to the oxidative environment. Despite the extensive investigations on the role of oxidative stress in collagen turnover, particularly related to various pathological events, in vitro studies utilizing direct cellular models are rather sparse [28]. Our recent study show that the oxidative environments inhibit the ability of ADMSCs to remodel adsorbed collagen type I not because they are harmful to the cells, but because of minute structural changes in the collagen molecules. To go more inside this phenomenon, here we describe the potential preventing role of bio-antioxidants. One of the most intensively studied antioxidants is the epigallocatechin-3-gallate (EGCG), the major polyphenolic compound of green tea (occupying about 30–40% of the dry weight), which is presumably the most frequently drunk beverage after water [29,30]. The phenol rings of EGCG act as electron scavengers of free radicals [31,32], thus inhibiting the formation of reactive oxygen species and reducing the harm caused by oxidative stress [33]. It has been shown, however, that EGCG effects are not only anti-oxidative, but may also also other biochemical routes, acting for example as an anti-inflammatory agent, with additional antiproteolytic, antiapoptotic, and antifibrotic functions [34,35], apart from its suggested protective effect in various cardiovascular and neurodegenerative diseases, diabetes, and certain cancers [34]. These diverse effects vary depending on cell type environment and levels of oxidants [36,37]. However, considering our recently observed effect of oxidation on stem cells remodeling of collagen type I [24], here we focus on the effect of EGCG on the ADMSCs remodeling in an oxidative environment, applying the same in vitro system.

## 2. Materials and Methods

### 2.1. Collagen Preparation

Mouse tail tendon collagen (MTC) was produced by acetic acid extraction and salting out with NaCl, as described elsewhere [38]. After centrifugation at 4000 rpm at 4 °C, the pellets were redissolved in 0.05 M acetic acid. The excess NaCl was removed by dialysis against 0.05 M acetic acid. All procedures were performed at 4 °C. The nearly monomolecular composition of a tail tendon, in which the collagen type I content approaches 100% of the total dry mass, was reasonably expected. The collagen concentration in the solutions was measured by modified Lowry assay [39] and from the optical absorbance at 220–230 nm [40].

### 2.2. Fluorescent Labeling of Collagen

The modified protocol of Doyle [41] was used for the FITC labelling of collagen obtained from murine tendons. MTC (2 mg/mL) was dissolved in 0.05 M borate buffer (pH 8), and 20 µg of FITC (from 1 mg/mL stock in DMSO) per 1 mg of protein was added and incubated at room temperature in the dark for 90 min. 0.05 M Tris buffer (pH 7.4) was used to stop the reaction, followed by extensive dialysis versus 0.05 M acetic acid, aiming to remove the excess FITC.

### 2.3. Collagen Oxidation Procedure

The MTC solution (2 mg/mL) was incubated in 0.05 M acetic acid, pH 4.3, with 50 µM FeCl_2_ and 5 mM H_2_O_2_ for 18 h at room temperature, as previously described [42]. The oxidant solutions were freshly prepared. 10 mM ethylenediaminetetraacetic acid (EDTA) was used to stop the reaction of oxidation, followed by intensive dialysis versus 0.05 M acetic acid, aiming to remove the excess of oxidants. Pretreatment with EGCG was performed via 30 min incubation of native collagen with 10 μM EGCG at room temperature before the oxidation procedure. The oxidized collagen (MTC-OXI) and EGCG pretreated (MTC-OXI/EGCG) were freshly prepared before all experiments.

### 2.4. Cells

Human ADMSCs of passage 1 was received from Tissue Bank BulGen using healthy volunteers undergoing liposuction with written consent. The cells were maintained in DMEM/F12 medium containing 1% GlutaMAX™, 1% Antibiotic-Antimycotic solution, and 10% Fetal Bovine Serum (FBS) all purchased from (Thermo Fisher Scientific, Life Technologies Corporation, Grand Island, NY, USA). Every two days the medium was replaced until the cells reached approximately 90% confluency and was then used for the experiments up to the 7th passage.

### 2.5. Morphological Studies

FITC labelled MTC (100 µg/mL), was coated (60 min) on (12 × 12) glass coverslips (ISOLAB Laborgeräte GmbH, Eschau, Germany) as solved in 0.05 M acetic acid. For the morphological studies, these collagen-coated slides were placed in 6-well standard tissues culture plates TPP® (Merck KGaA, Darmstadt, Germany). Then, the cells were seeded with 5 × 10^4^ cells/well ADMSCs at a final volume of 3 mL in serum-free medium and incubated for 2 h before the initial cell adhesion and overall cell morphology were monitored at phase contrast using inverted microscope Leica DM 2900 (Leica Microsystems, Mannheim, Germany). Next, 10% serum was added to each sample and the cells were cultivated up to 24 h, then washed 3× with PBS, fixed with 4% paraformaldehyde, and permeabilized with 0.5% Triton X-1000 for 5 min before fluorescence staining. Red fluorescent Rhodamine-Phalloidin (Invitrogen) was used (dilution 1:100) to visualize actin cytoskeleton, while cell nuclei were stained simultaneously by Hoechst 33,342 (Sigma-Aldrich/Merck KGaA, Darmstadt, Germany) in dilution 1:2000 for 30 min in PBS containing 10% albumin. Finally, the samples were mounted upside down on glass slides with Mowiol and viewed using the blue (for nuclei), red (for actin cytoskeleton), and green (for FITC-collagen) channels of an inverted fluorescent microscope (Olympus BX53, Upright Microscope, Olympus Corporation, Shinjuku Ku, Tokyo, Japan) at magnification 400× with objective (40×/0.50) UPlan FLN. A minimum of three representative images were obtained for each sample. The different colors were merged using the image processing software.

### 2.6. Measurement of Collagen Degradation by ADMSC

The collagen degradation assay by ADMSC is based on the de-quenching of fluorescently labelled protein, known as the FRET effect [43]. In the excess of dye during the labeling procedure, some of the label remains quenched (due to the high density of FITC molecules) and may de-quench (i.e., increase its fluorescence) upon the proteolytic degradation of the protein sample. Shortly afterwards, SensoPlate TM, 24 well Glass bottom, black plates (Lab Logistics Group GmbH, Meckenheim, Germany) were pre-coated with MTC, MTC-OXI, and MTC-OXI/EGCG solutions (100 µg/mL as above) in quadruplicated samples and washed 3 times with PBS before ADMSCs (1 × 10^4^ per well) were added in a final volume of 1 mL serum-free medium (assuring single protein adhesion of cells to collagen). After 2 h of incubation, 10% serum was added, and the cells were further cultured up to 24 h in a humidified CO_2_ incubator. Next, the supernatants were collected for the fluorescence measurement of released FITC labelled collagen, while the adsorbed (substratum associated) collagen was measured directly from the bottom of the plate (in 1 mL PBS) using Multimode Microplate Reader (Mithras LB 943, Berthold Technologies GmbH & Co. KG, Bad Wildbad, Germany) set at 485/525 nm. To estimate the particular effect of ADMSCs, control samples without cells (−cells) were processed in the same way. All experiments were quadruplicated. Fluorescence intensity is presented as relative photometric units (RPUs).

### 2.7. Quantitative Analysis of Raw Format Images by ImageJ

Qualitative data of fluorescent intensity were gathered via the measurement tool of Java build image postprocessing ImageJ and an additional plug-in FibrilTool was developed for ImageJ, measuring the anisotropy of the putative fibrillary structures [44]. The fluorescence intensity and the anisotropy of the fibrillary arrays were measured based on the raw format images of cells captured from at least three separate images under the same conditions. Several sequential steps were done: optimization, segmentation, analysis, and measurement steps. In the optimization module, pixel-based treatments were performed to highlight the regions of interest (ROIs) and allow the removal of artefacts. A default black and white threshold was used in the segmentation module. Images of equal size (W: 1600 px/H: 1200 px) were examined. The regions of interest (ROI) for the images of native MTC, MTC-OXI, and MTC-OXI/EGCG were delineated (a minimum of four ROIs for each collagen sample were examined). All measurements were performed at the green channel of the three colored images. The mean fluorescence intensity (MFI) and the anisotropy index (AI) were then calculated as the average of each ROI for the region of interest specified as: Outside the cell, Cellular region, and Central (nuclear/perinuclear) part. The MFI ratio inside/outside the cell was further calculated.

### 2.8. DSC Measurements

DSC measurements were performed using DASM4′s (Privalov, Biopribor, Pushchino, Russia) built-in high-sensitivity calorimeter with a cell volume of 0.47 mL. The samples of native MTC, MTC-OXI, and MTC-OXI/EGCG were prepared in 0.05 M acetic acid. The protein concentration was adjusted to 2 mg/mL. To prevent any degassing of the solution under study, a constant pressure of 2 atm was applied to the cells. The samples were heated at a scanning rate of 1.0 °C/min from 20 °C to 65 °C and were preceded by a baseline run with a buffer-filled cell. Each collagen solution was reheated after cooling from the first scan to evaluate the reversibility of the thermally induced transitions. The calorimetric curve corresponding to the second (reheating) scan was used as an instrumental baseline and was subtracted from the first scans, as collagen thermal denaturation is irreversible. The obtained excess heat capacity profiles were normalized to the protein concentration. The calorimetric data were analyzed using the Origin Pro 2018 software package.

### 2.9. Statistical Analysis

Data were analyzed using SPSS Statistics for Windows, version 23.0 (Armonk, NY, USA: IBM Corp). The quantitative results were obtained from at least four samples. Descriptive data were compared using Chi-square and Mann–Whitney U-tests. Nonparametric differences between groups were compared using the Friedman test; pairwise comparisons were achieved using the Dunn–Bonferroni post-hoc analysis. Data were expressed as mean ± standard deviation (SD). Differences with *p* < 0.05 were considered statistically significant and labelled with an asterisk in the figures.

### 2.10. Overall Design of the Experiments

ADMSC remodeling of adsorbed fluorescent collagen was studied by employing two approaches, morphological and enzymatic, comparing the native FITC-MTC and the oxidized collagen before and after EGCG pretreatment (Figure 1).

As shown in the diagram in Figure 1, the MTC was first labelled with FITC (FITC-MTC) to easily follow its fate upon adsorption. After the application of the protocol for oxidative modification as well as before or after EGCG pretreatment, the remodeling of collagen samples was investigated morphologically and via quantitative analysis of proteolysis. Morphological results were further quantified by morphometry.

## 3. Results

### 3.1. Morphological Study

In a preliminary study, we found morphologically that ADMSCs can reorganize the adsorbed FITC-labelled murine collagen (FITC-MTC), forming bright streaks of mechanically reorganized protein in a fibril-like pattern within 24 h of incubation (Figure 2). Signs for proteolytic remodeling of the adsorbed collagen were also observed (dark zones around some of the cells), presumably caused by peri-cellular proteolysis.

For more detailed morphological investigations, ADMSCs were cultured for 24 h on fluorescent FITC-MTC substrata (pre-coated glass coverslips) and then stained to simultaneously view the substratum adsorbed protein (green) and the cells stained for actin cytoskeleton (red) and nuclei (blue). The results are presented in Figure 3. In conjunction with the previous experiments, the adhering ADMSCs on regular collagen (left panels A, D, and G) tend to rearrange the underlining fluorescent layer, mechanically forming typical fibrillary assemblies mostly at the cells’ periphery. Note that characteristic dark zones of FITC-collagen removal in regions adjacent to the cells were also observed in native MTC samples (A, D, and G), suggesting the activation of peri-cellular proteolytic activity.

Conversely, on oxidized collagen, both mechanical and proteolytic activities tend to abrogate (images D–F), suggesting that oxidized collagen is hardly remodeled by ADMSCs, though the cells spread equally well on both substrata, showing similar polarized morphology (compare A–C). No morphological changes were observed for cells adhering to oxidized samples pretreated with EGCG (C,F,I). However, here the reorganization occurs differently, the cells tended to accumulate more collagen beneath, and the formation of some clusters located along the cell nuclei (Golgi localization) might be recognized, suggesting a partial internalization of the adsorbed collagen. Signs for adsorbed collagen degradation were missing.

No significant morphological difference was observed for the bare substrata, without cells. Some spontaneous fibrillary assembly of FITC-MTC might be observed, but it is rather independent of the oxidation (as shown in Figure 4) and presumably represents artefacts appearing from the spontaneous fibrillation of collagen in the coating solution.

### 3.2. Quantitative Morphometric Analysis

ImageJ software was further applied to better characterize the morphological observations. More specifically, a FibrilTool was used to quantify the overall fibrils’ organization [43] via measuring an anisotropy index (AI), and a Java built-in tool was further explored for measuring the fluorescence intensity (FI) in a given region of interest (ROI). For that purpose, ROIs of a similar size (±0.7%) were selected, namely: “Outside the cells”, the “Cellular region”, and the “Central (nuclear/perinuclear) part”, where both the AI and FI (in pixels) measured for the green, FITC-collagen channel only. The results are presented in Figure 5 and Figure 6 and are more detailed in the Appendix A.

Total anisotropy index (AI), for the whole sample, was measured for the bare collagen samples (−cells) and for samples with cells (+cells), as shown in Figure 5A and in the first two rows of the Appendix A. 

As expected, the bare MTC substrata showed very low AI, of about 0.07, suggesting a rather random collagen distribution. It was about twice lower (0.03) for the oxidized samples, while EGCG pretreatment partly “restored” the anisotropy index to about 0.05, though in the same low range. In the samples (+cells), however, the AI substantially increase, ranging between 0.21 for the native MTC collagen, lower for the oxidized sample (0.12), and lowest (0.08) for the EGCG pretreated one. Nevertheless, apart from the value for the bare EGCG pretreated substrata, here the anisotropy was substantially higher.

All this suggests that stem cells tend to reorganize all three substrata. To better characterize this increase of anisotropy in the presence of cells, a specific parameter ∆AI was introduced. For the native collagen samples, ∆AI was 0.143, while for oxidized and EGCG treated samples it drops to 0.088 and 0.023, respectively.

A similar trend was observed for the selected ROIs in the samples (+cells), as presented in Figure 5 panel B and in more detail in Appendix A. ROI outside cells again showed low anisotropy compared with the bare substrata, varying as 0.09, 0.020, and 0.060 for the native MTC, MTC-OXI, and MTC-OXI/EGCG, respectively. It sharply increased, however, in the ROIs cell region, rising to 0.383, 0.488, and 0.374, respectively, and confirming the significant reorganization of collagen by ADMSCs. It was surprisingly high for the oxidized MTC-OXI and MTC-OXI/EGCG, and thus did not well match the morphological observations, presumably due to the added values of ROIs from the central region (nuclear/perinuclear zone), showing a signal that was five times higher: 0.374 and 0.381 for the MTC-OXI and MTC-OXI/EGCG, respectively, and only 0.060 for the native MTC sample, suggesting a trend for different structuring the collagen within the perinuclear region.

The data in Figure 6 and Appendix A present the fluorescence intensity in different ROIs and suggest some accumulation of collagen in the ROI central region when compared to the ROI outside the cells.

Apart from the apparently higher signal for ROI (Inside cells) and ROI (Central region), compared to ROI (Outside cells), the FI was substantially higher for the nuclear regions of MTC-OXI, and particularly higher for the MTC-OXI/EGCG, reflecting well in the parameter “Ratio inside/outside cells” (Figure 6D, and the 5th row of Appendix A), amounting to 1.32 and 1.57, respectively, on the background of 1.17 for the native MTC samples. It has to be mentioned that the total area of the fluorescence measurement was approximately equal for all collagen samples, namely 1,904,820 for MTC, 1,878,228 for MTC-OXI, and 1,893,672 for MTC-OXI/EGCG, as presented in Appendix A.

### 3.3. Proteolytic Activity of ADMSCs

As described elsewhere [45], the measuring of proteolytic activity relies on the de-quenching of fluorescently labelled protein, known as the FRET effect [43]. This concept was proven for adsorbed proteins, including type 1 collagen, in our previous studies [45,46].

As shown in Figure 7, in the presence of ADMSCs, a substantial increase of the fluorescent signal (de-quenching) was observed in native MTC samples (left columns pare), suggesting a significant proteolytic activity of ADMSs toward adsorbed FITC-MTC collagen.

Data in Figure 7, however, show that this is not valid for the oxidized FITC-MTC-OXI and EGCG pretreated samples. Interestingly, in both cases, the fluorescent signal without cells (purple columns) showed higher fluorescence versus the signal in the presence of cells (blue columns). Although this difference was not significant (*p* > 0.05), we are prone to explain this trend with the “shadowing” effect of adhering cells. However, the fluorescence of EGCG pretreated samples was significantly higher when compared with the oxidized samples without pretreatment for the samples with cells and without cells.

### 3.4. DSC Thermograms and Thermodynamic Parameters of Collagen Samples

This experiment was performed to evaluate the putative impact of EGCG on the previously established change in the thermal stability of oxidized collagen observed by DSC [42], giving rise to an additional low-temperature pre-pick at 34.1 °C. More specifically, the aim here was to compare the thermal stability of MTC-OXI and MTC-OXI/EGCG. The heat capacity was measured as a function of temperature and presented as the transitional curves shown in, further used to calculate the melting temperature (T_m_), the total transition enthalpy (∆H total), and the half-widths of transition (Table 1).

As expected, the maximum heat absorption of native FITC-Col I was observed in the region of 39.6 °C (T_m2_) (Figure 8, Table 1). As a result of the oxidation, the thermogram splits into two well-resolved transitions with melting temperatures at 34.1 °C (T_m1_) and 39.6 °C (T_m2_), respectively (Table 1), which confirms our previous investigation [24,42].

As described elsewhere [24,42], the pre-transition of MTC-OXI sample results in the destabilization of the oxidized domains of collagen. However, no specific effect of EGCG pretreatment on the slope and the shape of thermograms was observed. According to the data in Table 1, the effect of EGCG pretreatment was also rather negligible. Nevertheless, the results presented demonstrate a small reduction in the total enthalpy of the transition (7.5%) and a slight change in the half-width of transition when compared to the oxidized collagen without pretreatment. Collectively, these data show that EGCG did not significantly change the oxidized collagen structure.

## 4. Discussion

Abnormal ECM remodeling upon oxidation has an important bio-medical aspect, as might be a sign of abrogated balance between the formation and degradation of ECM components, leading to pathological consequences [47]. The collagen turnover plays a significant role, encompassing the balance between its synthesis, organization, and degradation [48]. Although the common belief that the physiological role of its remodeling is to remove the excess or the altered collagen in the body [49], this structural protein also undergoes significant organization and reorganization by the adjacent cells, a process that plays a pivotal role in the structuring of ECM in tissues and organs [48]. Indeed, it often requires the action of proteases [49], which is crucial for its successful turnover. Proteolytic remodeling is mediated by several proteases, among which the matrix metalloproteinases (MMPs) are prominent, comprising a family of more than 23 zinc-dependent enzymes, which act not only by degrading the ECM components but also by processing the signaling molecules such as cytokines, chemokines, cell receptors, and growth factors [50].

Our recent study revealed that the adsorbed collagen Type I undergoes significant remodeling by stem cells, involving its mechanical reorganization in a fibril-like pattern, combined with the activation of cellular proteolytic machinery [38]. Likewise, we showed that both processes were abrogated in an oxidative environment due to the distinct structural alterations in collagen molecules [24]. The implication of stem cells in these studies, particularly ADMSCs, was an important issue considering their pivotal role in various regenerative routes. The effects of oxidative stress caused by ROS (or similar oxidative processes) on the repair of injured tissues are extensively studied, though still not well understood [51]. An important query here is whether this altered remodeling of collagen depends on the abrogated functioning of stem cells or is caused by intrinsic changes in the collagen molecules in an oxidative environment. In fact, we demonstrated that the abolished remodeling depends rather on the minute changes in the collagen structure, resulting in less susceptibility for proteases, than the altered cell functioning. The present study confirms this trend, but with collagen from another source, murine tendon, except for the previously used rat tail collagen [38].

The real novelty in this study comes from the implication of antioxidant EGCG (a well-known major constituent of the green tee), as we anticipate that it can block collagen oxidation, in this case via pre-treatment of the collagen solution. However, the DSC analysis shows that after pretreatment of collagen with EGCG (e.g., before oxidation) the denaturation curves were nearly identical to the oxidized sample, with the same slope, shape, and a very small reduction in heat enthalpy (roughly 7.5%), confirming discrete structural changes in the collagen molecule, but no “restoring” effect of the antioxidant. Thus, it turned out that the effect of EGCG is not so simple. The morphological investigations and the related quantifications (Figure 3, Figure 5, and Figure 6, and the Appendix A) confirmed that MTC underwent significant reorganization by ADMSCs, as evident from the sharp increase in anisotropy index (AI) of the whole sample in the presence of cells, as well as in the selected ROIs matching the cell regions. The AI, however, was surprisingly high for the oxidized MTC-OXI and MTC-OXI/EGCG, thus it notably did not confirm the morphological observations. We anticipate that the reason for this high anisotropy in the cell regions comes from the added values of ROIs (Central region), reflecting the nuclear and perinuclear zone of cells. They showed an anisotropy signal about five times higher, thus suggesting a trend for different structuring of collagen within the perinuclear region. In fact, distinct accumulation of FITC-collagen in this region was already observed morphologically and further confirmed with ImageJ morphometry analysis (Figure 6, and Appendix A), revealing that the fluorescence intensity (FI) was substantially higher for the nuclear regions of MTC-OXI, particularly in the EGCG pretreated sample. This implies the activation of other routes for intracellular processing of oxidized collagen by ADMSCs, presumably involving internalization and subsequent transcytosis. How it is supported by antioxidant EGCG, however, remains unclear. We hypothesize that the EGCG cannot prevent the oxidation of collagen (or does it only slightly) but may affect its subsequent processing by the cells. The morphological signs for the formation of clusters along the cell nuclei (Golgi localization) partly confirm this, suggesting that MSC may possess alternative routes for the removal of oxidized collagen, but that might be further supported by EGCG, for example, via endocytosis and vesicular trafficking ending with intracellular degradation of collagen. In the past few years, a great deal of understanding surrounding endosome functioning has been achieved. Proteins, following internalization at the cell surface, arrive at endosomes [52] for “sorting”, and afterwards some of them are delivered back to the cell surface via endosome-to-plasma membrane recycling, and other are sent to the trans-Golgi network via retrograde transport before reaching the degradative lysosomes [53]. We speculate that the oxidation may trigger endosome-to-plasma membrane recycling mechanism, assuring transcytosis of the oxidized collagen, while EGCG further activates retrograde transport to Golgi. The literature analysis shows that the effects of EGCG may involve direct interactions with plasma membrane components, including distinct proteins and phospholipids, unlocking intracellular signaling pathways. Thus, it may stimulate or inhibit various cellular processes, ranging from direct metabolic changes like lipogenesis and gluconeogenesis to apoptosis, autophagy, and mitochondrial and nuclear changes [33,54]. In addition, its chemical reactivity even makes it susceptible to generating reactive oxygen species (ROS) via auto oxidation, or to binding and crosslinking with other membrane proteins [34]. Therefore, EGCG could play a dual role, being both beneficial and harmful for the cells depending on concentration and the way of administration. In this case, we suppose a rather positive effect of EGCG, supporting the removal of inactive collagen molecules via its intracellular trafficking.

The enzymatic degradation is another rote for the remodeling of collagen, and it is known to be critical for in vivo collagen turnover. Based on previous studies, the degradation of adsorbed FITC-collagen might be easily quantified by measuring its de-quenching in the presence of ADMSCs proteases. As expected, the de-quenching here worked very well also, accompanied by the above-mentioned trend of mechanical remodeling in the presence of adhering ADMSCs. While the de-quenching was strongly expressed in the native collagen samples (Figure 7), it was significantly inhibited in oxidized collagen ones. The effect of the EGCG was surprising. Although the basic fluorescence background increased when compared to the oxidized samples, no de-quenching in the presence of ADMSCs was detected. On the contrary, some nonsignificant trends for diminished fluorescence in the presence of cells were observed, which we are prone to explain by optical shadowing of fluorescent signal from the adhering cells on the bottom of wells. Nevertheless, ADMSC-induced de-quenching in oxidized samples was absent, apart from the native collagen ones.

## 5. Conclusions

Collectively, this study confirms that ADMSCs hardly remodel oxidized collagen due to distinct structural changes in its molecules in an oxidative environment.

Collagen oxidation directs stem cells to activate alternative routes for collagen removal, such as internalization, processing, and transcytosis.

EGCG cannot block the oxidation of collagen but potentiates the removal of oxidized collagen via alternative routes.

## Figures and Tables

**Figure 1 polymers-14-03957-f001:**
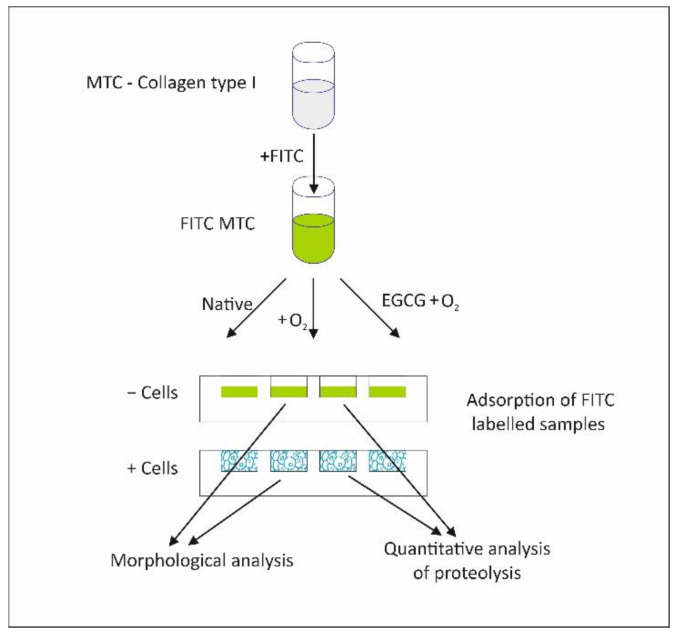
Overall experimental design.

**Figure 2 polymers-14-03957-f002:**
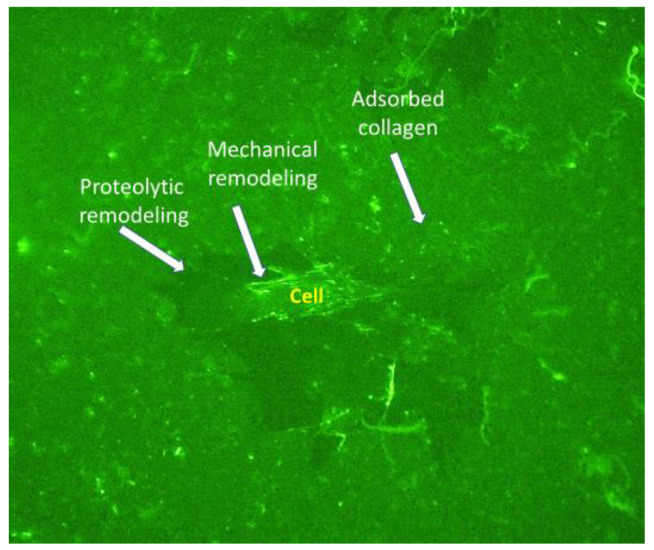
FITC labelled collagen remodeling by stem cells. ADSCS were cultured for 24 h on the fluorescent FITC-MTC substratum (green). The white arrows point to the places of adsorbed collagen with fibril-like reorganization and places of collagen removal caused by cellular proteolytic activity.

**Figure 3 polymers-14-03957-f003:**
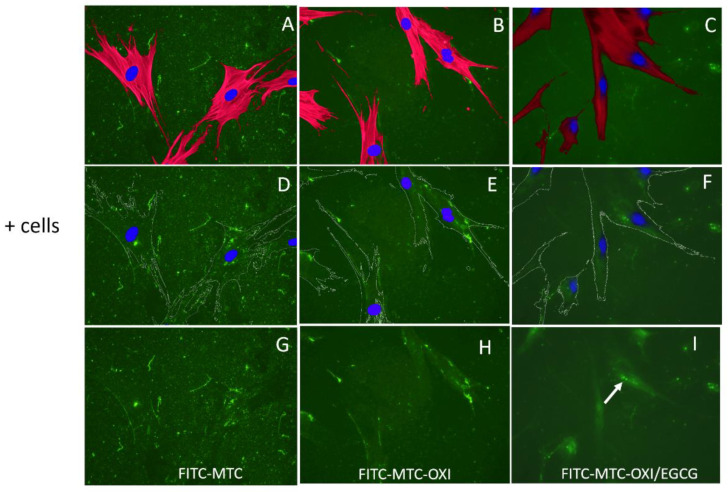
The overall morphology of ADSCS adhering to regular collagen (**A**,**D**,**G**), oxidized collagen (**B**,**E**,**H**), and oxidized collagen pretreated with EGCG (**C**,**F**,**I**). The top images (**A**–**C**) present the overall cell morphology viewed by the actin cytoskeleton (red) and cell nuclei (blue) on the green, fluorescent background of FITC collagen. The middle row (**D**–**F**) presents the substratum (green) with artificially superimposed cell contours and nuclei, while the bottom row (**G**–**I**) presents the collagen substratum only.

**Figure 4 polymers-14-03957-f004:**
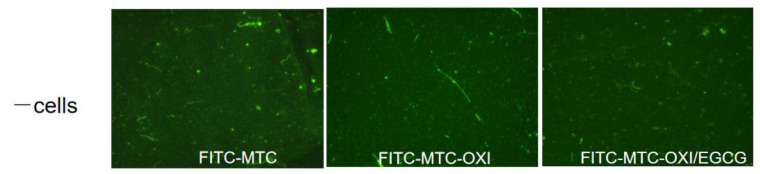
Spontaneous self-assembly of native FITC-MTC, oxidized (FITC-MTC-OXI), and oxidized collagen after EGCG pre-treatment (FITC-MTC-OXI/EGCG) samples.

**Figure 5 polymers-14-03957-f005:**
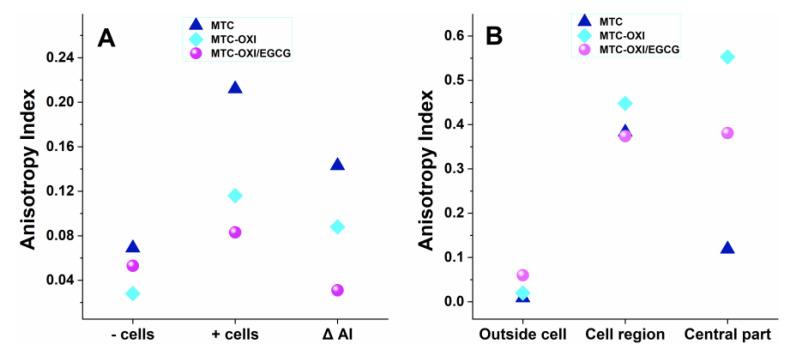
Panel (**A**). Anisotropy index of FITC labelled collagen samples, consisting of native (MTC), oxidized (MTC-OXI), and MTC-OXI pretreated with EGCG (MTC OXI/EGCG), measured for bare samples (−cells) and for samples (+cells). The change of anisotropy index by the cells is presented as ∆ AI. Panel (**B**). More detailed measurements of the same parameters were performed in the selected regions of interest (ROI): “Outside the cells”, the “Cellular region” and the “Central part” including the nuclear/perinuclear cell region.

**Figure 6 polymers-14-03957-f006:**
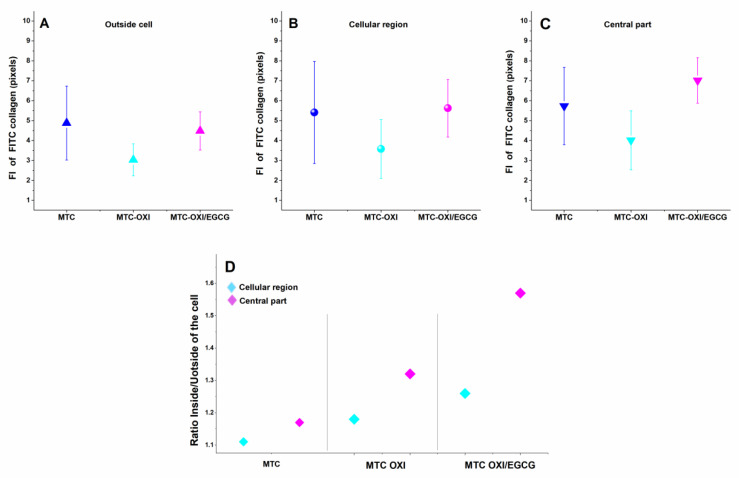
Fluorescence intensity (FI) in pixels of FITC collagen studied in different cell regions of interest (ROI): Outside the cells panel (**A**), the cellular region panel (**B**), and the central part—(**C**). The native (MTC), oxidized (MTC-OXI), and EGCG pretreated (MTC-OXI/EGCG) collagen samples were measured and compared. The FI ratio for Cellular region and the central part versus outside the cells, as calculated for the same collagen samples presented on panel (**D**).

**Figure 7 polymers-14-03957-f007:**
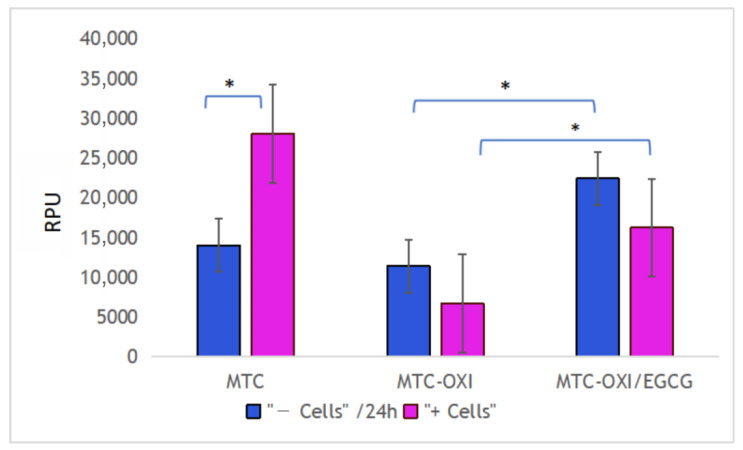
De-quenching activity of ADMSCs toward native (MTC), oxidized (MTC-OXI), and EGCG pretreated (MTC-OXI/EGCG) collagen samples, respectively. Fluorescence intensity is presented in relative photometric units (RPUs). The asterisk marks a significant difference with *p* < 0.05.

**Figure 8 polymers-14-03957-f008:**
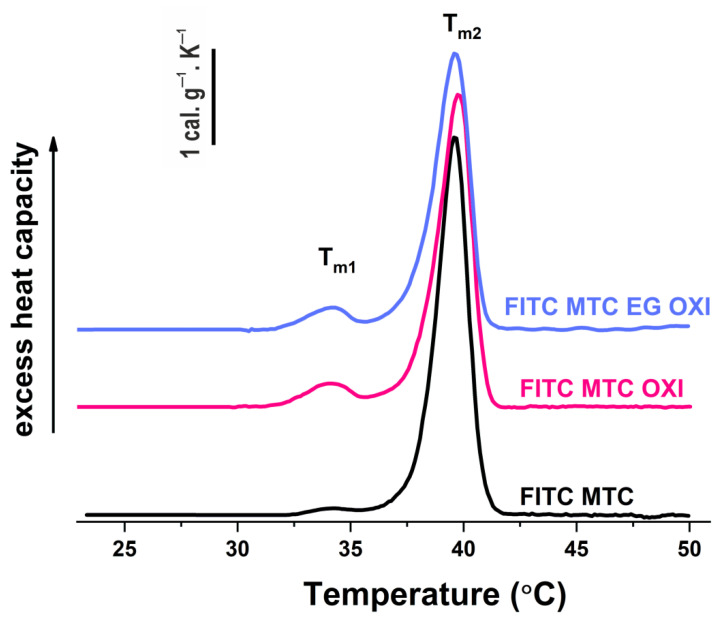
DSC thermograms of native FITC-MTC (black line), oxidized FITC-MTC-OXI (purple line), and EGCG pretreated FITC-MTC-OXI/EGCG (blue line) samples.

**Table 1 polymers-14-03957-t001:** Detailed thermodynamic parameters: transition temperature (T_m_), total calorimetric enthalpy (∆H total), and transition half-widths (T_m_ ½) obtained from DSC profiles of FITC-MTC, FITC-MTC-OXI, and FITC-MTC-OXI/EGCG.

Sample	T_m1_ (°C)	T_m2_ (°C)	∆H_total_ (cal g^−1^)	T_m2_ ½ (°C)
FITC-MTC	33.8	39.6	7. 2	1.61
FITC-MTC-OXI	34.1	39.7	7.00	1.80
FITC-MTC-OXI/EGCG	34.1	39.6	6.47	1.74

## Data Availability

Not applicable.

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
