# Peer review of "Morphological and Quantitative Evidence for Altered Mesenchymal Stem Cell Remodeling of Collagen in an Oxidative Environment—Peculiar Effect of Epigallocatechin-3-Gallate"

_polymers, 2022, doi:10.3390/polym14193957_

Round 1

Reviewer 1 Report

Comments to the Authors:

The Komsa-Penkova et al. entitled “Morphological and quantitative evidence for the altered mesenchymal stem cells remodeling of collagen in oxidative 3 environment – peculiar effect of epigalocatechin gallate”. Answering these comments are mandatory before publication.

1. Epigallocatechin-3-gallate (EGCG) is one of the most abundant and biologically active compound in green tea. Authors should using uniform name of EGCG in the manuscript.

2. The effects of EGCG are controversial, partly due to its poor instability in vitro in the previous study. According to overall experimental design (Figure 2), the MTC was first labeled with FITC (FITC-MTC), then treated with O2/EGCG+O2. However, oxidized tea polyphenols was obtained EGCG was treated with oxidant. Moreover, the morphological investigations were cultured for 24 h. Thus, oxidized tea polyphenols of EGCG was given under this condition. Please test for oxide production.

In addition, the experiment should add the group of EGCG.

3. Page 1, line 21, you should change " Epigallocathchin gallate (EGCG) " with " epigallocatechin-3-gallate (EGCG)".

4. For the first time, the ECM will use its full name in the manuscript.

5. --- In Manuscript, change " 40 minute " with "40 min".

6. There are some grammatical errors in the context of the manuscript. Please carefully revise the paper to increase language fluency.

7. The title name of 3.6 is needed modified.

8. The format of the table needs to be modified, such as Table 1, Table 2, and Table 3.

9. Please improve the quality of Figure 5.

10. The part of “CONCLUSION” needs to be resummarized.

11. There are many places the vocabulary mistakes add up to hamper the reading of the interesting manuscript. I will give only a few short examples:

   --- In Manuscript, change " 0.05M " with " 0.05 M".

Page 4, line 128, change " 18 hours " with " 18 h".

   Page 4, line 129, the EDTA was used its full name in the manuscript.

   Page 4, line 144, change " 12×12 " with " 18 h".

   Page 4, line 148, change " 22 " with " 2 h".

   Page 4, line 165, change " 1×104 per well " with " 1 × 104 per well ".

   Page 4, line 165, change " ml " with " mL ".

   Page 5, line 194, change " ml " with " mL ".

   Page 12, line 377, change " Fig.6 " with " Figure 6 ".

Author Response

Dear MRS Reviewer.

We appreciate the opportunity to resubmit our manuscript entitled “Morphological and quantitative evidence for altered mesenchymal stem cell remodeling of collagen in an oxidative environment – peculiar effect of epigallocatechin-3-gallate” to be considered for publication in “Polymers”,  Special issue “Polymer Materials in Biomedical Application II”.

We would also like to express our thanks to the reviewers for the very helpful comments. Accepting most of them we believe it has resulted in an improved version of the manuscript, which is uploaded alongside the document. We also enclose the point-by-point response to reviewers’ comments for your consideration.

Point-by-point answers to Comments and Suggestions for Authors
”  of reviewer 1.

The Komsa-Penkova et al. entitled “Morphological and quantitative evidence for the altered mesenchymal stem cells remodeling of collagen in oxidative environment – peculiar effect of epigallocatechin-3-gallate”. Answering these comments are mandatory before publication.

  1. Epigallocatechin-3-gallate (EGCG) is one of the most abundant and biologically active compound in green tea. Authors should using uniform name of EGCG in the manuscript.

Comment: We thank the reviewer for this note, the corresponding correction was done in the text: title, abstract, materials and methods (pages 1 and 3). 

  1. The effects of EGCG are controversial, partly due to its poor instability in vitro in the previous study. According to overall experimental design (Figure 2), the MTC was first labeled with FITC (FITC-MTC), then treated with O2/EGCG+O2. However, oxidized tea polyphenols was obtained EGCG was treated with oxidant. Moreover, the morphological investigations were cultured for 24 h. Thus, oxidized tea polyphenols of EGCG was given under this condition. Please test for oxide production.

In addition, the experiment should add the group of EGCG.

Comment: We agree with reviewer that EGCG effects are controversial, actually, this was the challenge of our investigation and the presented results. We made a small correction in the section “Materials and methods”, to better clarify, that FITC-collagen was first incubated with EGCG, then underwent oxidation. After these procedures collagen samples were extensively dialyzed to remove the excess of oxidants and potential ROS products. Finally the samples were coted on the coverslips (or TC wells) before the incubation with ADMSC for 24 hours. Thus, the cells were not in direct contact with EGCG during incubation. 

  1. Page 1, line 21, you should change " Epigallocathchin gallate (EGCG) " with " epigallocatechin-3-gallate (EGCG)".

Comment: We thank the reviewer for this suggestion, the corresponding corrections was done in the text.

  1. For the first time, the ECM will use its full name in the manuscript.

Comment: We thank for this note, the corresponding correction was done in the first line of introduction, page 1. 

  1. There are some grammatical errors in the context of the manuscript. Please carefully revise the paper to increase language fluency.

Comment: The manuscript was revised by a native English speaking reviewer and corrected when possible.

  1. The title name of 3.6 is needed modified.

Comment: Accepting this suggestion we modify the title 3.6 ( 3.4) as follows after the corrections:  

DSC thermograms and thermodynamic parameters of collagen samples (page 12).

  1. The format of the table needs to be modified, such as Table 1, Table 2, and Table 3.

Comment: We thank the reviewer for this note. In the revised version of the manuscript the Tables 1 and 2 are presented as Figures 5 and 6 according to the other reviewer’s suggestion. The format of Table 3 (Table 1 after the corrections), as well as the format of supplementary Tables A1 and A2 was updated accordingly.

  1. Please improve the quality of Figure 5.

Comment: Figure 5, in the revised version figure 7 was improved according to yours and other reviewer suggestion.      

  1. The part of “CONCLUSION” needs to be resummarized.

Comment: The conclusion was rephrased as follows:

Collectively, this study confirms that ADMSCs hardly remodel oxidized collagen due to distinct structural changes in its molecules in an oxidative environment.

Collagen oxidation directs stem cells to activating alternative routes for collagen removal like internalization, processing and transcytosis.

EGCG cannot block the oxidation of collagen, but potentiates the removal of oxidized collagen via alternative routes.

  1. There are many places the vocabulary mistakes add up to hamper the reading of the interesting manuscript. I will give only a few short examples:

Comment:  We acknowledge this remark of the reviewer. All reviewer suggestions were introduced in the manuscript accordingly.

   --- In Manuscript, change " 0.05M " with " 0.05 M".

Comment: The interval was introduced through the revised manuscript.

Page 4, line 128, change " 18 hours " with " 18 h".

Comment: The correction was introduced

   Page 4, line 129, the EDTA was used its full name in the manuscript.

Comment: The  full name  was introduced accordingly

   Page 4, line 144, change " 12×12 " with " 18 h". 

Comment: The expression 12×12 relates to the size of the glass slides -  coverslips, and it was was corrected to 12 × 12 with intervals

Page 4, line 148, change " 2h " with " 2 h".  

Comment: The interval was introduced

   Page 4, line 165, change " 1×104 per well " with " 1 × 104 per well ".

Comment: The intervals were introduced

   Page 4, line 165, change " ml " with " mL ".

   Page 5, line 194, change " ml " with " mL ".

Comment: SI nomenclature was used for millilitres, ml were corrected to  mL

   Page 12, line 377, change " Fig.6 " with " Figure 6 ".

The full name was introduced accordingly

Comment: We thank you for these detailed remarks. All these mistakes were corrected  in the text.  

Final remark: We thank the Reviewer 1 for the helpful suggestions and direct advises. We accepted them with pleasure and introduced, practically all of them, in the updated version of manuscript.

Sincerely yours

Prof. George Altankov

Prof. Regina Komsa-Penkova

9 September 2022

Reviewer 2 Report

This work investigates collagen remodeling by ADMSCs in a normal and oxidized environment, and whether EGCG pre-treatment has an effect. Morphological and enzymatic approaches were used to assess the FITC-labelled collagen, either native or oxidized +/- EGCG, on the substratum. The following suggestions may improve the presentation of the research.

1. In section 2.8, please describe the samples prepared for DSC.

2. The overall design of the experiments (3.1) and Figure 2 should be in materials and methods. Or, if a graphical abstract is required (one was not found in the manuscript), Figure 2 could be used for part of that.

3. Tables 1 and 2 would be better presented as graphs so that sample types and regions are more easily compared. Total area pixels in Table 2 can be mentioned in the text. Also, it is mentioned that values were increased. Please report the statistics and include error bars on the graphs for these values.

4. In section 3.5, the description of proteolytic activity measurement would be better placed in materials and methods.

5. In Figure 5, these graphs should be combined into 1 graph, "minus cells" bars should be to the left of "plus cells" bars, and statistical significance should be shown with asterisks and or lines. Additionally, there is no explanation for what the "-02" represents in the right-positioned graph.

Author Response

Dear MRS Reviewer,

We appreciate the opportunity to resubmit our manuscript entitled “Morphological and quantitative evidence for altered mesenchymal stem cell remodeling of collagen in an oxidative environment – peculiar effect of epigallocatechin-3-gallate” to be considered for publication in “Polymers”,  Special issue “Polymer Materials in Biomedical Application II”.

We would also like to express our thanks to you for the very helpful comments. Accepting most of them we believe it has resulted in an improved version of the manuscript, which is uploaded alongside the document. We also enclose the point-by-point response to reviewers’ comments for your consideration.

Point-by-point answers to Comments and Suggestions for Authors
”  of reviewer 2.

This work investigates collagen remodeling by ADMSCs in a normal and oxidized environment, and whether EGCG pre-treatment has an effect. Morphological and enzymatic approaches were used to assess the FITC-labelled collagen, either native or oxidized +/- EGCG, on the substratum. The following suggestions may improve the presentation of the research.

  1. In section 2.8, please describe the samples prepared for DSC.

Comment: We thank the reviewer for this note, the corresponding description was introduced in section 2.8.

  1. The overall design of the experiments (3.1) and Figure 2 should be in materials and methods. Or, if a graphical abstract is required (one was not found in the manuscript), Figure 2 could be used for part of that.

Comment: We appreciate this suggestion. In the revised version the section “Overall design of the experiments is placed in the Materials and Methods section, subsection 2.10. The Graphical abstract contains a part of this information but with another design. We are sorry that this information did not reach you.

  1. Tables 1 and 2 would be better presented as graphs so that sample types and regions are more easily compared. The total area pixels in Table 2 can be mentioned in the text. Also, it is mentioned that values were increased. Please report the statistics and include error bars on the graphs for these values.

Comment: We acknowledge this comment. In the revised version of the manuscript, the content of Tables 1  and 2 are presented as Figures 5 and 6, while the full information is presented as supplementary Tables A1 and A2. The total area in pixels is presented in the text on page 13 and in the supplementary Table A2.

  1. In section 3.5, the description of proteolytic activity measurement would be better placed in materials and methods.

Comment: We thank for this suggestion, the description was placed in the Materials and methods section, subsection 2.6.

  1. In Figure 5, these graphs should be combined into 1 graph, "minus cells" bars should be to the left of "plus cells" bars, and statistical significance should be shown with asterisks and or lines. Additionally, there is no explanation for what the "-02" represents in the right-positioned graph.

Comment: We acknowledge this remark. Figure 5, (Figure 7 in the revised version) was redone accordingly; the A and B graphs were combined into one graph and asterisks and lines were added where realizable to point to the statistical significance and further mentioned in the text.

Final remark: We thank Reviewer 2 for the helpful suggestions and direct advises. We accepted them with pleasure and introduced, practically all of them, in the updated version of the manuscript.

Sincerely yours

Prof. George Altankov

Prof. Regina Komsa-Penkova

9 September 2022

Reviewer 3 Report

Dear Authors,

I appreciated the quality and the completeness of your work. 

It is developed and presented with the correct skills requested. Probably your recent publication on International Journal of Molecular Science has been a useful path to follow. 

Best regards

Author Response

Dear MRS Reviewer.

We appreciate the opportunity to resubmit our manuscript entitled “Morphological and quantitative evidence for altered mesenchymal stem cell remodeling of collagen in an oxidative environment – peculiar effect of epigallocatechin-3-gallate” to be considered for publication in “Polymers”,  Special issue “Polymer Materials in Biomedical Application II”.

We would like to express our gratitude and appreciation for the high evaluation of our manuscript.

Sincerely yours

Prof. George Altankov

Prof. Regina Komsa-Penkova

9 September 2022
